# Development and Evaluation of an Index to Measure the Ability to Get Vaccinated for COVID-19

**DOI:** 10.3390/vaccines11020342

**Published:** 2023-02-03

**Authors:** William D. Evans, Jeffrey B. Bingenheimer, Michael Long, Khadi Ndiaye, Dante Donati, Nandan M. Rao, Selinam Akaba, Bailey Hoglin

**Affiliations:** 1Milken Institute School of Public Health, The George Washington University, Washington, DC 20052, USA; 2School of Business, Columbia University, New York, NY 10027, USA; 3Virtual Lab LLC, Corvallis, OR 97330, USA

**Keywords:** COVID-19, vaccine hesitancy, motivation, opportunity, ability, social media, social norms

## Abstract

The COVID-19 pandemic has been an historic challenge to public health, and to behavior change programs. There have been challenges in promoting vaccination in LMICs, including Nigeria. One important hypothesis deserving consideration is the ability to obtain vaccination as a potential barrier to vaccination uptake. The MOA (motivation, opportunity, and ability) framework, as illustrated by multiple theories such as COM-B, EAST, and the Fogg model, is a primary theoretical basis for the evaluation of this ability as a factor in vaccination uptake. There is little research on measuring the ability to get vaccinated in LMICs, including on the role of all of the MOA framework. The aim of this study was to develop and evaluate an ability factors index measured through social media-based data collected in Nigeria in late 2021 and early 2022. We present findings from an online survey of 8574 Nigerians and highlight new social media-based data collection techniques in this research. This study found that a new ability factors index comprising 12 items was associated with vaccine uptake independent of measures capturing other components of the MOA framework. This index may serve as a valuable research instrument for future studies. We conclude that a person’s perceived ability to get vaccinated, measured by a newly validated index, is related to vaccination uptake and hesitancy, and that more research should be conducted in this area.

## 1. Introduction

Behavioral theory is a crucial component of effective public health interventions. There is a growing body of evidence that theory-based interventions are more successful in health behavior change programs compared to interventions lacking theoretical underpinnings [1,2]. A recent systematic review found that this is especially true in international development programs in low- and middle-income countries (LMIC), where relatively straightforward and pragmatic frameworks can be applied despite resource constraints and other implementation barriers [3]. One such framework is the motivation, opportunity, and ability (MOA) framework. First developed by MacInnis and colleagues [4] in the field of marketing, the framework was reconceptualized for health-related and social behaviors by Rothschild [5] and has been applied to numerous health-related behaviors [6]. In this manuscript, we describe the development of an index to measure the ability component of the MOA framework in the context of efforts to promote uptake of COVID-19 vaccinations.

The use of theories for health promotion and efforts to change unhealthy behaviors is rooted in an understanding that health and social development problems do not exist in isolation. They are a function of interacting factors—sociocultural, economic and geographic—at different levels, for example, individual, family, and community (including institutional factors), that impact personal agency and individual choices and decisions [1]. Therefore, health behaviors are critically intersectional in that they cannot be understood based on one factor but rather multiple factors that merge in diverse ways in connection with micro and macro environments, race, ethnicity, gender, biology, socioeconomic status, and psychosocial factors.

This is especially true with regard to access to health resources and inequalities, as exemplified by the COVID-19 pandemic [7]. For a newly developed technology such as a vaccine against COVID-19, there is a hypothetical diffusion curve ranging from the early adopters who actively seek vaccination, to those who are unsure or hesitant, to those who actively resist and/or are actively opposed to vaccination [8]. For example, COVID-19 demand creation among the early adopters (i.e., the enthusiastic) is relatively easy as they will come forward without much persuasive effort [9]. In a situation with adequate vaccine supply, however, those who are hesitant will come to represent more and more of the unvaccinated over time. As the hesitant come to the front of the queue, there will be a need for more targeted approaches that speak to the concerns held by these groups.

There is currently limited evidence for perceived constraints (or perceived lack of ability) contributing to vaccine hesitancy [3]. Research is needed on potential ability barriers and theory-based interventions to overcome them. While structural barriers may represent ability barriers in the population (e.g., lack of vaccine stock, long wait times or distances to vaccination centers), social and behavior change interventions can be effective at tackling perceived barriers that drive hesitancy and inhibit uptake (e.g., the misperception that getting a vaccine is time-consuming or costly) and removing friction in the process (e.g., showing where the nearest vaccination facility is located) [9].

Since not all groups possess the motivation (i.e., attitudes and beliefs), opportunity (i.e., situational conditions), and ability (i.e., task knowledge) to modify behaviors [10], some research offers a conceptual framework for guiding and regulating public health behaviors through tools available in education, marketing, and law [5]. This framework views motivation, opportunity, and ability as key categories of variables influencing behavioral choice. It posits that perceptions of self-interest and trade-offs present in the marketplace of choices constrain what interventions can do to maximize societal-level health and well-being [9].

Within the MOA framework, the ability factor may be considered to have six dimensions: time, money, physical effort, mental effort, social norms, and routine [3]. The current study attempts to operationalize these dimensions and to develop and validate an index to measure perceived ability to obtain COVID-19 vaccination. We hypothesized that such a valid index of ability in the context of COVID-19 vaccine uptake might be associated with other variables that predict vaccination, including vaccine hesitancy and the five Cs of vaccination uptake, identified in previous research [11,12], but would exert a unique influence on vaccine uptake over and above these other variables. This paper reports on survey research to collect and analyze the new ability index.

## 2. Methods

### 2.1. Design

The current study was part of a larger evaluation of the impact of social media campaigns to promote COVID-19 vaccination among health care workers and others in their social environment in Nigeria. The evaluation employed mixed methods, and comprises a quantitative study, conducted through social media-based surveys, a qualitative study of stakeholders, and a cost-effectiveness study. Within the quantitative study, we collected baseline data from 8574 participants through a Facebook-based survey in December 2021. As part of this research, we developed and fielded a 12-item ability index to a sub-sample of 781 participants, who were randomly selected from the main sample. Data were collected in 6 Nigerian states that received the social media intervention promoting COVID-19 vaccination, and compared to data collected in the other 31 non-treatment states. The primary endpoint was self-reported vaccination uptake. Additional, secondary outcomes of interest included vaccine hesitancy, and the 5 Cs of vaccination (confidence, complacency, constraints, calculation, and collective responsibility), and social norms related to vaccination [13,14,15]. This study was reviewed and approved as exempt by The George Washington University Institutional Research Board (IRB) in August 2021, and received Nigerian IRB approval from the National Primary Health Care Development Agency (NPHCDA) in September 2021.

### 2.2. Measures and Item Development

The ability factors index was developed based on a conceptual model of vaccine uptake that was based on the MOA framework and diffusion of innovation theory [5,8,12]. In this conceptual model, the motivation component of the MOA framework is captured by four of the five Cs of vaccine hesitancy [13]—confidence, complacency, calculation, and collective responsibility—and by perceived descriptive and injunctive norms related to vaccine uptake. The opportunity component of the MOA framework is captured by the remaining C of vaccine hesitancy: convenience. Following previous research on the dimensions of ability [C]—time, money, physical effort, mental effort, social norms, and routine—we developed and pilot tested a 12-item index with study participants in Nigeria recruited via social media. The 12 items are shown in the results section below, and consisted of multiple items across the 6 noted dimensions of ability. This brief pilot test indicated the items in the index were well-understood, feasible, and usable for future exploratory research.

As described below, the ability factors index was fielded as a supplement to a larger, social media-based survey designed to evaluate the COVID-19 vaccination promotion campaign in Nigeria. Other measures included the five Cs scale [10], a social norms for vaccination scale adapted from previous social norms research [15,16], measures of social and mass media use, and vaccination status.

### 2.3. Data Collection

Survey data for the current study, and the overall evaluation of which it is a part, were collected on a social media-based research platform called Virtual Lab (https://vlab.digital/ accessed on 2 January 2023). The survey instrument was designed to measure the previously mentioned outcomes, as well as key elements in the campaign theory of change (ToC), which included COVID-19 vaccination-related knowledge, attitudes, beliefs, intentions, and behaviors. The ToC was designed to identify mediating variables, which include social norms, role modeling, and cultural beliefs that are theoretically related to vaccine intentions and vaccination behavior.

Virtual Lab is an open-source software platform that uses digital advertising to recruit a custom, stratified pool of respondents. In this case, the study stratified by health care worker status, and we sought to obtain 50% of our total sample in this group, with the rest being general Nigerian population. Targeted participants had to have a Facebook account, and they received recruitment advertising promoting a study on COVID-19 vaccination. By clicking through the ad, potential participants then engaged with a chatbot that works with Facebook Messenger (FM). A series of FM posts screened and obtained consent from participants following the IRB-approved protocol. Eligible participants were offered an incentive of NGN 400 (about USD 1) in mobile phone credits to complete a questionnaire consisting of 40 items, each delivered as an individual FM message. Data were captured and stored in a secure Virtual Lab database.

As part of this nationwide data collection, we also developed and fielded an ‘ability factors’ module. The module followed previous research on ability within the MOA framework and identified 6 dimensions of the ability factor: time, money, physical effort, mental effort, social norms, and routine [3,17]. These 6 dimensions represent potential ability barriers to taking action, in this case to receive COVID-19 vaccination. The specific survey measures, all asked on a 5-point agreement scale from strongly agree to strongly disagree, are shown in the results section below.

We randomly selected, with 10% probability, a sub-sample of respondents from the main FM-administered survey to receive an additional NGN 400 and complete the ability factors module. Participants who agreed received these additional questions by Facebook Messenger and then finished the survey.

Individuals who consented to participate were sent questionnaire items one at a time via that application. For most items, participants were prompted simply to tap on the response option of their choice to answer that item. After answering each item, participants received the next item within a few seconds, and this process was repeated through the end of the questionnaire. Responses were collected in an electronic database and exported as a .csv file for analysis. The ability index items were administered only to a randomly selected 10% subsample of the overall study sample, which resulted in a total of 781 recruited participants.

### 2.4. Data Analysis

The analytic approach for this study was shaped by our conceptualization of ability as an induced rather than a latent variable. Rather than assuming the existence of a unidimensional underlying latent construct that is the primary driver of responses to the individual indicators, we regard ability as being the outcome of the amalgamation of its components, which are captured by the indicators. This has two primary analytic implications. First, as others have argued [18,19,20,21], internal consistency as measured, for example, by Cronbach’s alpha or evaluated through exploratory or confirmatory factor analysis is not a requirement for indices as it is for scales. The inter-item correlations that underly Cronbach’s alpha, the eigenvalues in exploratory factor analysis, and the factor loadings in both exploratory and confirmatory factor analysis arise in latent variable models because the indictors are assumed and intended to share a common cause in the target construct. For indices, however, the relationship between indicators and construct is reversed. With the indicators functioning as causes of the construct rather than the reverse, there is no reason to suppose that the indicators will be correlated with one another.

All of our main analyses are, therefore, geared toward validity rather than internal consistency [18]. We conduct two sets of validity analyses at the item level. First, we examine the variability in responses to each of the twelve items, looking for items in which the responses are very heavily skewed toward one end of the response scale or the other. Such items have constrained variability and may not provide much useful information about individual differences. Second, we examine bivariate associations between each of the ability indicators and our main criterion variable: vaccination status at baseline. We do this via a series of logistic regression models with vaccination status as the dependent variable, each containing one of the ability items as the sole independent variable.

The second implication of our conceptualization of ability as an induced rather than latent variable is a focus on how the components represented by the twelve indicators combine to create ability. We consider two alternatives. In the first, we define ability as the simple unweighted sum of the twelve components or indicators. In the second, we define it as a weighted sum of the indicators. To derive the weights, we randomly split the sample into two subsamples of approximately equal size. We use the first subsample (which we term the developmental sample) to develop the weights. Specifically, we estimate a logistic regression model in which vaccination status at baseline is the dependent variable, and the twelve ability indicators serve as the independent variables. We use the coefficients from this model as the weights in deriving the second version of the index.

Next, we examine the validity of both versions of the index in two ways. First, in the full sample, we examine correlations between each version of the index and five indicators of vaccine hesitancy and five indicators of social norms related to COVID-19 vaccination. The motivation for these analyses is that, while ability may be correlated to some extent with some of these other variables, those correlations should be, at most, moderate in magnitude, suggesting that the ability index measures something distinct from vaccine hesitancy and social norms [19]. Next, we examine the concurrent validity of both versions of the index by using it to predict baseline vaccination status via logistic regression analyses in the holdout sample, first without and then with the five vaccine hesitancy and five social norms indicators as control variables. Third, we examine the predictive validity of both versions of the index by using it, first without and then with the vaccine hesitancy and social norms indicators as control variables, in logistic regression models predicting vaccine uptake at follow-up among the subset of participants who reported being unvaccinated at baseline. In all of these index-level validity analyses, we compare (albeit not with statistical hypothesis tests) the performance of the two versions of the index, unweighted and weighted.

Although, as discussed above, we prefer to conceptualize ability as an induced rather than latent variable, we do nevertheless carry out some conventional internal consistency and factor analyses. Details of the methods and results of these analyses are presented as Appendix A. Stata version 17 (College Station, TX, USA) was used for all analyses.

## 3. Results

Table 1 shows the composition of the 10% subsample to which the ability indicators were administered. The sample is relatively young, with over 60% of participants being under 30 years of age. The sample is also well educated, with over 60% of participants having a bachelor’s degree or higher. Just over half of the sample are not employed in the health sector, and the health sector workers are fairly evenly divided among several categories of health sector workers. The sample contains more men than women, and the modal category of religion is non-Catholic Christians. Over 60% of the sample was already vaccinated at baseline.

### 3.1. Validity of the Ability Items

The distributions of responses to the twelve ability items are shown in Table 2. There is substantial variability in the responses to all twelve of the items. For five of the items (1, 4, 7, 9, and 10), the modal response was “Disagree”, and for another five (2, 3, 5, 6, and 11) the modal response was “Agree”. The remaining two items (8 and 12) show bimodal distributions, with peaks at both “Agree” and “Disagree”. None of the items are so skewed that either of the extremes (“Strongly agree” or “Strongly disagree”) was the modal response. For all subsequent analyses, items 1, 4, 7, 8, 9, 10, and 12 were reverse coded so that higher values consistently represent greater ability.

Table 3 shows bivariate associations between each of the 12 items and baseline vaccination status in the form of crude odds ratios derived from logistic regression models. All associations are in the hypothesized positive direction, with point estimates suggesting that a one-unit increase in responses to the item is associated with increases of between 9% (for item 10) and 62% (for item 8) in the odds of being vaccinated. All but one of the odds ratios (for item 10) are statistically significantly different from zero at the conventional a = 0.05 level.

### 3.2. Derivation of the Ability Index

Table 4 shows the exponentiated coefficients (adjusted odds ratios) from the multivariable logistic regression model in which the twelve ability indicators serve as independent variables and baseline vaccination status in the developmental subsample. Interestingly, only two of the adjusted odds ratios are statistically significantly different from the null (items 6 and 8). Moreover, three of the items (items 5, 7, and 10) had adjusted odds ratios below one, although these were not statistically significantly different from zero. The two versions of the ability index—one defined as the simple sum of the twelve indicators and the other defined as the weighted sum with the logistic regression coefficients (the unexponentiated versions of the adjusted odds ratios from Table 4) serving as the weights—were highly but not perfectly correlated (Pearson correlation 0.84) in the full sample.

### 3.3. Validity of the Ability Index

Correlations between these two versions of the ability index and (a) five indicators of vaccine hesitancy and (b) five indicators of social norms related to COVID-19 vaccination are shown in Table 5. While some of the correlations are substantial and statistically significant, none are very large. Moreover, point estimates for the unweighted version of the ability index are consistently in the same direction, but somewhat larger in absolute value than in the weighted version.

Table 6 shows associations between the two versions of the ability index and baseline vaccination status in the holdout sample, without and then with controls for items measuring vaccine hesitancy and social norms around COVID-19 vaccination, as assessed via logistic regression. Panel A uses the unweighted version of the index, and Panel B uses the weighted version. In each panel, Model 0 has the ability index as the sole independent variable, Models 1 through 10 control for the vaccine hesitancy and social norms items one at a time, and Model 11 controls for all hesitancy and norms items simultaneously. In both panels, the crude odds ratio is large and statistically significant (2.20 for the unweighted version and 2.46 for the weighted version). Controlling for individual hesitancy or norms generally moves these odds ratios toward the null, but more so for some control variables (e.g., convenience) than others (e.g., social norms among people close to you). For both versions of the ability index, the odds ratios remain statistically significant and in the hypothesized direction for all specifications of the model, including those in which all ten vaccine hesitancy and social norms indicators are included as control variables. Additionally, across specifications, the estimated odds ratios for the weighted version of the ability index are larger than those for the unweighted version.

Similarly, associations between the two versions of the ability index and vaccination status at follow-up among those who reported being unvaccinated at baseline, without and then with controls for items measuring vaccine hesitancy and social norms, are shown in Table 7. For both the unweighted and weighted versions of the index, the odds ratios across model specifications are all in the hypothesized direction, with point estimates ranging from 1.23 to 1.30 for the unweighted index, and from 1.31 to 1.39 for the weighted index across specifications. None of these crude or adjusted odds ratios, however, are statistically significantly different at the conventional *p* = 0.05 level.

## 4. Discussion

We developed and tested a 12-item ability factors index to operationalize the construct of perceived ability to get vaccinated following the MOA framework [4,5]. This ability factors index was also informed by diffusion of innovations and the five Cs model of vaccine uptake [8,12]. The resulting items were pilot tested, and the findings indicate that they are feasible and acceptable to participants. We then proceeded to use the index in a quasi-experimental evaluation study of a COVID-19 vaccination promotion social media campaign through a nationwide social media-based survey in Nigeria.

In this study, we developed and tested an ability factors index and its applicability to analysis of determinants of COVID-19 vaccination, including the five Cs, vaccine hesitancy, and social norms. In answer to RQ1, we succeeded in identifying a valid ability factors index and tested it with the hypothesized COVID-19 vaccination determinants.

The ability factors index was associated with several determinants of COVID-19 vaccination. Thus, we confirmed H1 in several respects. First, we found numerous small but statistically significant associations between both the weighted and unweighted versions of the ability factors index and the five Cs and social norms items. Thus, ability, as measured in our index, may be a useful predictor of change in these determinants of vaccination, and future research should investigate these associations in longitudinal and experimental designs.

Second, in multivariate models to assess the associations between the ability index and vaccination status in the holdout sample, without and then with controls for items measuring vaccine hesitancy and social norms around COVID-19 vaccination, we found significant associations between ability and determinants of vaccination. The associations involved in our previous models between ability and the 5 Cs and social norms held, for the most part, in the multivariate models, suggesting that the relationships observed between ability and these determinants of vaccination were robust.

Based on these findings, we suggest that understanding theory-based models of vaccine uptake, and in particular the role of perceived and objective ability to vaccinate, is crucial for effective vaccination promotion interventions [22]. The MOA framework is a useful and practitioner-friendly theoretical model to use in designing vaccination interventions, and ability is an important factor in this model [23,24]. The index developed in this study is a potentially valuable tool in future research [25,26]. Future studies should include rigorous evaluation of the role of ability in predicting vaccine uptake, including studies that specifically deliver vaccine demand creation campaigns to increase participants’ perceived sense of ability, and thereby promote vaccination [26,27,28].

For example, future studies could examine the relationship between the ability index and specific factors associated with vaccine hesitancy [29]. These include perceptions of vaccine availability, safety, efficacy, and other benefits (such as protecting family, friends, and community). Using sampling approaches available through the social media data collection approach used in this study, future RCTs can identify vaccine-hesitant populations that may be susceptible to persuasion based on increasing their perceived ability to vaccinate, as measured by the ability index [30].

### Limitations of the Current Research

This study was conducted on a relatively small sub-sample of a larger evaluation sample. Research with a larger sample, with longitudinal follow-up to examine changes in the ability factors index over time, would expand on the current study findings. The social media-based sample was collected by convenience on the Facebook platform, and is thus not nationally representative of Nigerians. Finally, the lack of a randomized experimental design in this study limits the generalizability of the findings.

## 5. Conclusions

Ability is a key construct in the MOA framework, a widely used theoretical model that is practitioner-focused and has broad applicability in development programming in LMICs [3]. We successfully developed and tested a multi-dimensional ability factors index and found that it was statistically associated with other outcomes of interest related to vaccine hesitancy and vaccine uptake. The ability factors index has potential as a measurement and analytical tool in future vaccine promotion and other social and behavior change interventions. Future research should apply and further test this promising index for its applicability in predicting ability as a determinant of behavior.

## Figures and Tables

**Table 1 vaccines-11-00342-t001:** Sociodemographic characteristics of the sample.

Age in Years	*n*	%
18–29	469	60.1
30–39	230	29.5
40–49	60	7.7
50–59	18	2.3
60+	4	0.5
Education		
Primary or secondary school	121	15.5
Diploma	180	23.1
Bachelors	349	44.7
Masters	78	10.0
PhD	17	2.2
Other	36	4.6
Gender		
Man	502	64.3
Woman	267	34.2
Prefer not to say	12	1.5
Employment		
Not a health sector worker	405	51.9
Nurse/midwife	70	9.0
Community health worker	64	8.2
Laboratory staff	56	7.2
Medical doctor	55	7.0
Pharmacist	35	4.5
PPMV/chemists	16	2.1
Other health sector worker	80	10.2
Religion		
Christian (not Catholic)	391	50.1
Muslim	219	28.0
Catholic	152	19.5
Other	19	2.4
Vaccination Status		
Not vaccinated	297	38.0
Vaccinated	484	62.0

**Table 2 vaccines-11-00342-t002:** Distribution of responses to ability index items (*n* = 781).

	Strongly Agree	Agree	Neutral	Disagree	Strongly Disagree
1. My family and household responsibilities make it difficult for me to find time to get a COVID-19 vaccine.	17	80	108	334	242
	(2.2)	(10.2)	(13.8)	(42.8)	(31.0)
2. I can leave my work long enough to get the COVID-19 vaccine.	123	317	160	120	61
	(15.8)	(40.6)	(20.5)	(15.4)	(7.8)
3. The COVID-19 vaccine is affordable to me.	121	329	152	114	65
	(15.5)	(42.1)	(19.5)	(14.6)	(8.3)
4. I would lose income by taking time to get the COVID-19 vaccine.	22	131	129	313	186
	(2.8)	(16.8)	(16.5)	(40.1)	(23.8)
5. My supervisor would support my getting the COVID-19 vaccine.	208	324	148	46	55
	(26.6)	(41.5)	(19.0)	(5.9)	(7.0)
6. The COVID-19 vaccine is conveniently available to me.	102	340	129	158	52
	(13.1)	(43.5)	(16.5)	(20.2)	(6.7)
7. I have to travel a long way to get the COVID-19 vaccine.	43	182	97	323	136
	(5.5)	(23.3)	(12.4)	(41.4)	(17.4)
8. I am worried that side effects of the COVID-19 vaccine would make it difficult for me to fulfill my work or family responsibilities.	49	219	156	233	124
	(6.3)	(28.0)	(20.0)	(29.8)	(15.9)
9. The decision to get the COVID-19 vaccine is difficult.	46	190	126	294	125
	(5.9)	(24.3)	(16.1)	(37.6)	(16.0)
10. I have a spouse/partner who would not approve of me getting a COVID-19 vaccination.	40	87	135	315	204
	(5.1)	(11.1)	(17.3)	(40.3)	(26.1)
11. I know where to go to get the COVID-19 vaccine.	157	367	77	119	61
	(20.1)	(47.0)	(9.9)	(15.2)	(7.8)
12. It would require a big change in my daily routine to get the COVID-19 vaccine.	45	224	149	269	94
	(5.8)	(28.7)	(19.1)	(34.4)	(12.0)

**Table 3 vaccines-11-00342-t003:** Associations between ability index items and baseline vaccination (*n* = 781).

Twelve Ability Index Items	OR	(95% C.I.)
1. My family and household responsibilities make it difficult for me to find time to get a COVID-19 vaccine.	1.43	(1.24–1.65) ***
2. I can leave my work long enough to get the COVID-19 vaccine.	1.34	(1.18–1.52) ***
3. The COVID-19 vaccine is affordable to me.	1.54	(1.35–1.75) ***
4. I would lose income by taking time to get the COVID-19 vaccine.	1.24	(1.09–1.41) **
5. My supervisor would support my getting the COVID-19 vaccine.	1.44	(1.26–1.65) ***
6. The COVID-19 vaccine is conveniently available to me.	1.92	(1.67–2.21) ***
7. I have to travel a long way to get the COVID-19 vaccine.	1.31	(1.16–1.49) ***
8. I am worried that side effects of the COVID-19 vaccine would make it difficult for me to fulfill my work or family responsibilities.	1.62	(1.42–1.84) ***
9. The decision to get the COVID-19 vaccine is difficult.	1.46	(1.29–1.66) ***
10. I have a spouse/partner who would not approve of me getting a COVID-19 vaccination.	1.09	(0.96–1.24)
11. I know where to go to get the COVID-19 vaccine.	1.57	(1.39–1.79) ***
12. It would require a big change in my daily routine to get the COVID-19 vaccine.	1.27	(1.12–1.45) ***

*** = *p* < 0.001; ** = *p* < 0.01.

**Table 4 vaccines-11-00342-t004:** Adjusted odds ratios (AORs) and 95% confidence intervals (CIs) from the multiple logistic regression model with ability index items predicting baseline vaccination status in the development sample (*n* = 386).

	AOR	(95% C.I.)
1. My family and household responsibilities make it difficult for me to find time to get a COVID-19 vaccine.	1.23	(0.97–1.56)
2. I can leave my work long enough to get the COVID-19 vaccine.	1.20	(0.95–1.52)
3. The COVID-19 vaccine is affordable to me.	1.15	(0.90–1.47)
4. I would lose income by taking time to get the COVID-19 vaccine.	1.01	(0.79–1.29)
5. My supervisor would support my getting the COVID-19 vaccine.	0.95	(0.73–1.22)
6. The COVID-19 vaccine is conveniently available to me.	1.85	(1.41–2.44) ***
7. I have to travel a long way to get the COVID-19 vaccine.	0.94	(0.74–1.19)
8. I am worried that side effects of the COVID-19 vaccine would make it difficult for me to fulfill my work or family responsibilities.	1.40	(1.11–1.75) **
9. The decision to get the COVID-19 vaccine is difficult.	1.14	(0.91–1.44)
10. I have a spouse/partner who would not approve of me getting a COVID-19 vaccination.	0.88	(0.70–1.10)
11. I know where to go to get the COVID-19 vaccine.	1.12	(0.89–1.41)
12. It would require a big change in my daily routine to get the COVID-19 vaccine.	1.04	(0.82–1.33)

*** = *p* < 0.001; ** = *p* < 0.01.

**Table 5 vaccines-11-00342-t005:** Correlations between the ability indices and indicators of the five Cs and social norms in the holdout sample (*n* = 395).

	Version 1	Version 2
I am confident that COVID-19 vaccines are safe and effective.	0.18 ***	0.15 **
Vaccination against COVID-19 is unnecessary.	−0.32 ***	−0.22 ***
Everyday stress prevents me from getting a COVID-19 vaccine.	−0.36 ***	−0.25 ***
When I think about getting vaccinated against COVID-19, I weigh the benefits and risks to make the best decision possible.	−0.03	−0.03
When everyone is vaccinated against COVID-19, I don’t have to get vaccinated too.	−0.40 ***	−0.25 ***
Your friends think it is important for everyone to get a COVID-19 vaccine.	0.28 ***	0.22 ***
Your family members think it is important for everyone to get a COVID-19 vaccine.	0.38 ***	0.33 ***
Of the people close to you, what proportion of them would want you to get the COVID-19 vaccine?	0.10 *	0.07
How many people in Nigeria do you think will get the COVID-19 vaccine when it becomes available?	0.03	−0.00
How many people who work in healthcare in Nigeria do you think will get the COVID-19 vaccine when it becomes available?	0.13 *	0.08

NOTE: Version 1 of the ability index is the unweighted sum of the twelve individual items, and Version 2 is the weighted sum of the items with the weights being the logistic regression coefficients presented in Table 4. *** = *p* < 0.001; ** = *p* < 0.01; * = *p* < 0.05.

**Table 6 vaccines-11-00342-t006:** Associations of the ability index with vaccination in the holdout sample, without and with control for five Cs and social norms indicators (*n* = 395).

	Model 0	Model 1	Model 2	Model 3	Model 4	Model 5	Model 6	Model 7	Model 8	Model 9	Model 10	Model 11
A. Unweighted Index												
Ability Index	2.20 ***	2.18 ***	2.20 ***	2.00 ***	2.20 ***	2.13 ***	2.20 ***	2.09 ***	2.20 ***	2.20 ***	2.21 ***	1.92 ***
Confidence		1.05										1.06
Complacency			1.00									1.10
Lack of Convenience				0.74 ***								0.72 ***
Calculation					0.86							0.83
Collective Responsibility						0.92						1.01
Friends							1.00					0.94
Family								1.16				1.27
People Close to You									0.99			1.99
Nigerians										1.02		1.03
Health Care Workers											0.96	0.94
B. Weighted Index												
Ability Index	2.46 ***	2.44 ***	2.43 ***	2.35 ***	2.47 ***	2.37 ***	2.45 ***	2.35 ***	2.46 ***	2.46 ***	2.46 ***	2.43 ***
Confidence		1.05										1.06
Complacency			0.95									1.09
Lack of Convenience				0.69 **								0.70 **
Calculation					0.85							0.84
Collective Responsibility						0.84						0.94
Friends							1.02					0.95
Family								1.16				1.24
People Close to You									1.00			0.99
Nigerians										1.05		1.05
Health Care Workers											0.98	0.96

*** = *p* < 0.001; ** = *p* < 0.01.

**Table 7 vaccines-11-00342-t007:** Prospective associations of the ability index with vaccination, without and with control for five Cs and social norms indicators (*n* = 196).

	Model 0	Model 1	Model 2	Model 3	Model 4	Model 5	Model 6	Model 7	Model 8	Model 9	Model 10	Model 11
A. Unweighted Index	1.28	1.24	1.26	1.27	1.28	1.27	1.25	1.28	1.23	1.29	1.30	1.29
Ability Index		1.14										1.14
Confidence			0.68 *									0.60 **
Complacency				0.88								0.88
Lack of Convenience					1.13							1.12
Calculation						0.95						1.26
Collective Responsibility							1.14					1.11
Friends								1.00				0.84
Family									1.21			1.15
People Close to You										0.96		0.89
Nigerians											1.15	1.20
Health Care Workers												
B. Weighted Index												
Ability Index	1.38	1.35	1.34	1.38	1.38	1.37	1.36	1.38	1.31	1.39	1.39	1.32
Confidence		1.14										1.15
Complacency			0.68 *									0.61 *
Lack of Convenience				0.87								0.87
Calculation					1.13							1.13
Collective Responsibility						0.95						1.24
Friends							1.14					1.12
Family								1.02				0.86
People Close to You									1.19			1.14
Nigerians										0.95		0.89
Health Care Workers											1.15	1.19

** = *p* < 0.01; * = *p* < 0.05.

## Data Availability

Data and supporting reported results may be obtained by written request to the corresponding author.

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
