# Peer review of "Development and Evaluation of an Index to Measure the Ability to Get Vaccinated for COVID-19"

_vaccines, 2023, doi:10.3390/vaccines11020342_

Round 1

Reviewer 1 Report

The manuscript focuses on developing and evaluating an index to measure the ability for covid-19 vaccination. The development of the index seems appropriate.

If the indicators function as causes of the construct (line 176), why did you describe using an oblique rotation in the factor analysis you conducted (results not presented)?  I am trying to understand conceptually what you did.

Is there a citation to support your conjecture that "those correlations should be at most moderate in magnitude" (Line201)?

Is there a way to present the results in fewer tables or reduce the information in the tables?  I assume in Table 2, the values in parentheses are the percentages?  This should be noted.

The authors should remove the word proved and replace it with another word, line 309

The paragraph in the discussion can be removed, lines 313-321. This is a repeat of what you described earlier in the manuscript.

Author Response

Response to reviewer #1:

The manuscript focuses on developing and evaluating an index to measure the ability for covid-19 vaccination. The development of the index seems appropriate.

AUTHOR RESPONSE: THANK YOU, WE APPRECIATE THE FEEDBACK.

If the indicators function as causes of the construct (line 176), why did you describe using an oblique rotation in the factor analysis you conducted (results not presented)?  I am trying to understand conceptually what you did.

AUTHOR RESPONSE: In a sense, we see question from the reviewer as consisting of two parts. The first is: If the indicators function as causes of the construct, why do factor analysis at all? And the second is: Given that you’re doing a factor analysis, why use an oblique rotation rather than an orthogonal one? Here we address the second question first. We preferred to use oblique rotation because orthogonal rotation involves an assumption that we thought was likely to be unrealistic: namely that, if there were multiple factors, those factors would be uncorrelated with each other. The oblique rotation approach, in contrast, is more flexible, allowing the correlation between the latent factors to be estimated as part of the factor analysis and rotation. Returning to the first part of the question, although we believe it makes sense to conceptualize ability as an induced rather than a latent variable (as explained at length on lines 164 to 177 of the manuscript), we recognize that the latent variable approach and its corresponding analytic approaches including internal consistency analysis (e.g., Cronbach’s alpha) and factor analysis, is more widely used and may make more sense to some investigators for this application. We realize, however, that the paper as a whole and the methods section in particular is quite long, and in the resubmission we have therefore moved those analyses and the results thereof into supplementary materials.

Is there a citation to support your conjecture that "those correlations should be at most moderate in magnitude" (Line201)?

AUTHOR RESPONSE: This is based upon the chapter on validity in the DeVellis and Thorpe book, which is reference number 19 in the paper. This issue is specifically discussed on page 85 of that book.

Is there a way to present the results in fewer tables or reduce the information in the tables?  I assume in Table 2, the values in parentheses are the percentages?  This should be noted.

AUTHOR RESPONSE: WE FEEL THAT THE TABLES ARE APPROPRIATE TO FULLY CONVEY THE INFORMATION FROM THIS STUDY. WE HAVE UPDATED TABLE 2 AS REQUESTED.

The authors should remove the word proved and replace it with another word, line 309

AUTHOR RESPONSE: WE UPDATED THE TEXT TO SAY “FINDINGS INDICATE.”

The paragraph in the discussion can be removed, lines 313-321. This is a repeat of what you described earlier in the manuscript.

AUTHOR RESPONSE: THANK YOU, WE EDITED THE PARAGRAPH TO REMOVE REDUNDANCIES.

Reviewer 2 Report

The manuscript is poorly organized and lengthy.

1.      Introduction & method section: generally, lack of inherent logic of the construct of the scale and vaccine hesitancy.

a.      Reference 3 “Within the MOA framework, the Ability factor may be considered to have 6 dimensions: Time, money, physical effort, mental effort, social norms, and routine [3]line 77” and then to method section line 146 the same ability factor:” identified 6 dimensions of the ability factor: Time, money, physical effort, 146 mental effort, social norms, and routine [3, 17]”has two reference. No difference was spotted between them and why 2 different citations.

b.      The same redundant but inconsistent expression appears in Line99 “and the 5 Cs of vaccination (confidence, complacency, constraints, calculation, and collective responsibility)”and line 111 “The opportunity component of the MOA framework is captured by the remaining C of vaccine hesitancy: convenience”. Where does convenience come from as the remaining C of vaccine hesitancy?

c.      Line19,” all asked on a 5-point agreement scale from strongly agree to strongly disagree, are shown in the results section below.” Should be 5 point-Likert scale.

d.      Data analysis is wordy and lacks readability. But the authors failed to provide the random selection methods of the 10% participants for sub-population.

2.      Results:

Table 4 presented coefficients of multiple logistic regression hard to follow and interpret. Please employ adjusted OR and 95% CI.

Table 6 and 7 are unnecessary, no mention to put weighted index section,

3.discussion and conclusion focus on ability factors but the result section(especially Table 6 and 7) is beyond that and makes readers confused.

Author Response

Response to reviewer #2

The manuscript is poorly organized and lengthy.

  1. Introduction & method section: generally, lack of inherent logic of the construct of the scale and vaccine hesitancy.

AUTHOR RESPONSE: WE THANK THE REVIEWER FOR THE FEEDBACK. WE HAVE EDITED THE INTRODUCTION IN AN ATTEMPT TO IMPROVE THE FLOW AND HAVE REDUCED LENGTH.

  1. Reference 3 “Within the MOA framework, the Ability factor may be considered to have 6 dimensions: Time, money, physical effort, mental effort, social norms, and routine [3]line 77” and then to method section line 146 the same ability factor:” identified 6 dimensions of the ability factor: Time, money, physical effort, 146 mental effort, social norms, and routine [3, 17]”has two reference. No difference was spotted between them and why 2 different citations.

AUTHOR RESPONSE: THE 2 REFERENCES WERE PROVIDED TO SHOW THAT THERE ARE MULTIPLE SOURCES OF SUPPORT FOR THE MEASUREMENT APPROACH USED.

  1. The same redundant but inconsistent expression appears in Line99 “and the 5 Cs of vaccination (confidence, complacency, constraints, calculation, and collective responsibility)”and line 111 “The opportunity component of the MOA framework is captured by the remaining C of vaccine hesitancy: convenience”. Where does convenience come from as the remaining C of vaccine hesitancy?

AUTHOR RESPONSE: THIS SECTION REFERS TO A PREVIOUSLY PUBLISHED SCALE, THE 5 Cs. CONVENINECE IS ONE OF THE 5 Cs IN THAT PUBLISHED SCALE.

  1. Line19,” all asked on a 5-point agreement scale from strongly agree to strongly disagree, are shown in the results section below.” Should be 5 point-Likert scale.

AUTHOR RESPONSE: WE HAVE MADE THE REQUESTED REVISION.

  1. Data analysis is wordy and lacks readability. But the authors failed to provide the random selection methods of the 10% participants for sub-population.

AUTHOR RESPONSE: WE HAVE EDITED THIS SECTION.

  1. Results:

Table 4 presented coefficients of multiple logistic regression hard to follow and interpret. Please employ adjusted OR and 95% CI.

AUTHOR RESPONSE: WE EDITED TABLE 4.

Table 6 and 7 are unnecessary, no mention to put weighted index section,

3.discussion and conclusion focus on ability factors but the result section(especially Table 6 and 7) is beyond that and makes readers confused.

AUTHOR RESPONSE: WE THANK THE REVIEWER BUT RESPECTFULLY SUBMIT THAT TABLES 6-7 ARE NEEDED TO PROVIDE FULL INFORMATION FROM OUR STUDY.

Reviewer 3 Report

1)     “Ability factor index” should be defined in one sentence somewhere prior using it.

2)     Which method was used to collect the data from social media has not been mentioned (questionnaire/interview etc.?).

3)     Age group of young adults has not been defined.

4)     Can we elaborate little what parameters have been covered under ability factor index as it is becoming non-specific in abstract.

5)     Proof reading is required as some sentences are not correctly framed in the main text.

6)     We can add some statistics on “percentage hesitancy for covid 19 vaccine” shown by people of different countries in introduction section.

7)     Vaccine hesitancy is complex and context-specific, varying between the time, place, and type of vaccine, we can add bit elaboration regarding these also in introduction if authors agree, only then.

8)     Study was conducted on health workers! It can be mentioned under the objective of the study in the last paragraph of introduction section.

9)     Sampling method is not clear. How the subsampling was done. How the participants were recruited earlier. Is the total number of participants (i.e., 8574) are the participants who responded to the survey at first time? If yes, then after that what sampling method has been used for the sub sample selection, this needs to be specified. What is basis for the decision of the sub-sample size from main sample size? Was it based on any existing studies or decided out of convenience. This can be mentioned.

10)  Baseline data is collected on what parameters need to be mentioned.

11)  P value should be given in table 2 for better understanding of significant difference. In the tables, individual p values can be included in the table in each row and boldened whenever it is significant.

12)  Mean, SD should be given for participant’s age.

13)  How attitude towards vaccines and political orientation was assessed? Any conversion factor or scale was used to determine vaccine hesitancy?

14)  How reliability of the tools used was checked? Pearson r value was significant or not? Need to mention.

15)  Normal distribution of data was checked before analysis? In the form of skewness, kurtosis etc.

16)  Data analysis is too long and conceptual. The section can be restricted only key points regarding the statistical tests used, and how it was calculated (software or other means)

17)  Results can also be explained better by using correlation between different factors and hesitancy. Conclusion can also include how this tool can be applied in the current as well as other scenarios of vaccinations.

18)  Number of tables given in the MS can be reduced.

19)  Discussion can have more information regarding ability index and how it helps for predicting attitude towards covid related beliefs.

20)  There are a few recent publications which refers to vaccine hesitancy and other learnings from vaccination programs, which need to be cited such as :

 Lahariya C, Paruthi P & Bhattacharya M. “How a new health intervention affects the health system? Learnings from pentavalent vaccine introduction in India.” Indian J Pediatr 2016; 83: 294-9.

Author Response

Response to reviewer #3:

  • “Ability factor index” should be defined in one sentence somewhere prior using it.

AUTHOR RESPONSE: WE ADDED A BRIEF DEFINITION AS REQUESTED.

  • Which method was used to collect the data from social media has not been mentioned (questionnaire/interview etc.?).

AUTHOR RESPONSE: WE NOTE THAT THE VIRTUAL LAB PLATFORM WAS USED. THIS WAS IN THE MANUSCRIPT, BUT WE HAVE ADDED A CLARIFICATION.

  • Age group of young adults has not been defined.

AUTHOR RESPONSE: THE STUDY DOES NOT SPECIFICALLY REFER TO YOUNG ADULTS.

  • Can we elaborate little what parameters have been covered under ability factor index as it is becoming non-specific in abstract.

AUTHOR RESPONSE: WE EDITED THE ABSTRACT AND NOTE THAT THE PARAMETERS OF ABILITY ARE DESCRIBED IN THE METHODS SECTION.

  • Proof reading is required as some sentences are not correctly framed in the main text.

AUTHOR RESPONSE: THANK YOU, WE HAVE THOROUGHLY COPYEDITED THE MANUSCRIPT.

  • We can add some statistics on “percentage hesitancy for covid 19 vaccine” shown by people of different countries in introduction section.

AUTHOR RESPONSE: WE HAVE BEEN ASKED BY OTHER REVIEWERS TO SHORTEN THE INTRODUCTION, AND THUS HAVE NOT ADDED DATA FROM COUNTRIES OTHER THAN NIGERIA.

  • Vaccine hesitancy is complex and context-specific, varying between the time, place, and type of vaccine, we can add bit elaboration regarding these also in introduction if authors agree, only then.

AUTHOR RESPONSE: WE HAVE EDITED THE DISCUSSION OF VACCINE HESITANCY WITHIN SPACE CONSTRAINTS.

  • Study was conducted on health workers! It can be mentioned under the objective of the study in the last paragraph of introduction section.

AUTHOR RESPONSE: WE HAVE CLARIFIED THE INCLUSION/EXCLUSION CRITERIA.

  • Sampling method is not clear. How the subsampling was done. How the participants were recruited earlier. Is the total number of participants (i.e., 8574) are the participants who responded to the survey at first time? If yes, then after that what sampling method has been used for the sub sample selection, this needs to be specified. What is basis for the decision of the sub-sample size from main sample size? Was it based on any existing studies or decided out of convenience. This can be mentioned.

AUTHOR RESPONSE: WE HAVE ADDED DETAILS ON THE SAMPLING APPROACH WITHIN SPACE CONSTRAINTS.

  • Baseline data is collected on what parameters need to be mentioned.

AUTHOR RESPONSE: WE HAVE ADDED DETAILS ON THE SAMPLING METHODS.

  • P value should be given in table 2 for better understanding of significant difference. In the tables, individual p values can be included in the table in each row and boldened whenever it is significant.

AUTHOR RESPONSE: WE HAVE EDITED TABLE 2.

  • Mean, SD should be given for participant’s age.

AUTHOR RESPONSE: WE ADDED THIS INFORMATION.

  • How attitude towards vaccines and political orientation was assessed? Any conversion factor or scale was used to determine vaccine hesitancy?

AUTHOR RESPONSE: THIS STUDY DID NOT ADDRESS POLITICAL ORIENTATION.

  • How reliability of the tools used was checked? Pearson r value was significant or not? Need to mention.

AUTHOR RESPONSE: It is not clear to us what tools the reviewer is referring to here, or to what Pearson correlations. If the Pearson’s correlations the reviewer is referring to are those in Table 4, however, we note that their statistical significance is indicated by the number of asterisks next to each coefficient.

  • Normal distribution of data was checked before analysis? In the form of skewness, kurtosis etc.

AUTHOR RESPONSE: All of the analyses we present either (a) do not involve any assumption about the normality of the variables, or (b) are quite robust to minor deviations from normality when implemented in large samples like this one. Note in particular that the twelve items that comprise the ability index use an ordinal response scale, and thus cannot be truly normally distributed, but that the frequency distributions shown in Table 2 reveal that the responses are only moderately skewed.

  • Data analysis is too long and conceptual. The section can be restricted only key points regarding the statistical tests used, and how it was calculated (software or other means)

AUTHOR RESPONSE: Thank you, we have edited the data analysis section in an effort to make it shorter and easier to follow. Note in particular that we have removed nearly the entire last paragraph, which now appears as part of a Supplemental Material document.

  • Results can also be explained better by using correlation between different factors and hesitancy. Conclusion can also include how this tool can be applied in the current as well as other scenarios of vaccinations.

AUTHOR RESPONSE: Thank you, we have edited the results section to make the explanations clearer, paying specific attention to the parts that focus on the correlations between the two versions of the ability index and the five indicators of vaccine hesitancy in Table 5.

  • Number of tables given in the MS can be reduced.

AUTHOR RESPONSE: WE RESPECTFULLY KEPT ALL THE TABLES AS THEY PROVIDE COMPLETE INFORMATION FROM THE STUDY.

  • Discussion can have more information regarding ability index and how it helps for predicting attitude towards covid related beliefs.

AUTHOR RESPONSE: WE ADDED FURTHER DETAILS ON HOW THE ABILITY INDEX MAY BE USED IN FUTURE RESEARCH.

20)  There are a few recent publications which refers to vaccine hesitancy and other learnings from vaccination programs, which need to be cited such as :

 Lahariya C, Paruthi P & Bhattacharya M. “How a new health intervention affects the health system? Learnings from pentavalent vaccine introduction in India.” Indian J Pediatr 2016; 83: 294-9.

AUTHOR RESPONSE: WE ADDED THIS REFERENCE. THANK YOU.

Round 2

Reviewer 2 Report

I don't think the authors seriously address the reviewer's comments. 

Author Response

We have made additional revisions to Table 4, as requested. Also, we made further revisions to the Introduction and have double checked language for consistency and accuracy. We thank the reviewer for the 2nd review.
